# Voices from the Rainbow: Exploring Participants’ Experiences in an Online LGBTIQ+-Affirmative Psychoeducation Program

**DOI:** 10.3390/healthcare13020115

**Published:** 2025-01-09

**Authors:** Ruth A. Ancín-Nicolás, João Carlos Collado, Miguel A. Lopez-Sáez, António-José Gonzalez

**Affiliations:** 1Department of Psychology, Rey Juan Carlos University, Avda/Atenas, s/n, 28922 Alcorcón, Spain; ruth.ancin@urjc.es (R.A.A.-N.); miguel.lopez.saez@urjc.es (M.A.L.-S.); 2AppsyCI, Ispa—Instituto Universitário, Rua Jardim do Tabaco, 34, 1140-041 Lisboa, Portugal; gonzalez@ispa.pt

**Keywords:** affirmative therapy, LGBTQ, minority stress model, psychoeducation

## Abstract

**Background/Objectives**: This article highlights the importance of addressing the mental health of LGBTQ people, specifically through psychoeducation/intervention programs. The primary objective was to understand the effects of participating in an online affirmative program. The theoretical frameworks of the minority stress model and the concept of decompensation were used to understand and address disparities with the general population. **Methods**: A detailed protocol was designed to understand the participants’ experiences. An analysis of the results was carried out using a content analysis of the data collected from a convenience sample of participants from two groups. **Results**: The findings show that the participants’ experiences were satisfactory, especially with developing coping skills and changing their attitudes towards homophobia, also showing improvements in mental health indicators. **Conclusions**: This study concludes that online LGBTQ-affirmative psychoeducation programs can be positive in enhancing the mental health and well-being of the LGBTQ population. The research highlights the importance of extending these programs to LGBTQ family members. It is also important to acknowledge the specificities of each subgroup.

## 1. Introduction

Over the past few decades, mental health indicators within the lesbian, gay, bisexual, trans*, and queer (LGBTQ) population have received increasing scientific attention [1]. This burgeoning field of study has highlighted significant disparities in mental health outcomes between the general population and sexual and gender minorities [2,3]. These disparities highlight that LGBTQ individuals often experience poorer mental health when compared to their heterosexual and cisgender counterparts [4]. It is crucial to note that identifying as LGBTQ does not inherently predispose one to poorer mental health; rather, contextual stressors and multifactorial discrimination play a pivotal role in this phenomenon [5,6,7]. Consequently, there is a pressing need to deepen our understanding of discrimination and stigma and their profound effects on the mental health of this population.

LGBTQ individuals face unique forms of discrimination, encapsulated by the “iii” acronym: insult, isolation, and invisibility [8]. Discrimination can manifest implicitly, through the non-inclusion or non-recognition of an LGBTQ person’s identity, or explicitly, via verbal or physical violence [9]. Such violence and insults not only affect direct victims, but also instill fear and insecurity within the broader LGBTQ community, prompting many to conceal their identities to avoid further victimization [9]. Kitzinger characterizes these acts of violence as non-random, targeting individuals as symbols of the groups to which they belong or are perceived to belong [10]. The radicalization of the whole of Europe, including Portugal, puts LGBTQ people at greater risk due to the hate speech of the far right against this community [11]. This increases the psychosocial stress of this population [12], especially bisexual or trans people [13,14]. Violence and discrimination occur in many contexts. The healthcare system is one of them. Portuguese LGBTQI+ people have also documented blatant cases of heterosexism and heteronormativity in the healthcare system [15,16]. Furthermore, external stigmatization and rejection are also internalized. They are often a product of internalized homophobia, stemming from the adoption of negative societal attitudes, which leads to self-devaluation and internal conflict [17,18].

Meyer delineates distal and proximal components within the minority stress model to explain the adding of stressors that minorities face [19,20]. The distal component encompasses external stressors, such as interpersonal discrimination, victimization, hate crimes, and microaggressions [21]. The proximal component involves the internalization of stigma through cognitive and affective processes, including internalized LGBTQ-phobia and the anticipation of stigma, leading to anxiety and constant self-awareness for the concealment of one’s sexual orientation or gender identity [21].

Moreover, the theory of decompensation clarifies how people with minority identities need to use compensatory strategies to mitigate the negative impacts of everyday discrimination, stigmatization, and ideological conflicts that challenge their right to exist. When systems that support these strategies are overloaded, people suffer from psychological anguish. The cumulative impact of microaggressions and the healing potential of compensatory mechanisms, including the empowering aspects of LGBTQ culture, are highlighted by decompensation theory. However, there is a threshold beyond which compensation is no longer feasible, due to the resource depletion caused by ongoing adversity [22].

The consequences of such discrimination are manifold, including a high risk of depression, post-traumatic disorder [23], suicidality and self-harm [24,25], substance use [26], and the consumption of cognitive resources in maintaining invisibility, which could otherwise be used for personal, academic, or professional development [23]. Persistent exposure to bullying and negativity fosters expectations of rejection, which in turn promote avoidance behaviors such as isolation and invisibility, perpetuating a vicious cycle [27].

Despite all the violence suffered, Meyer’s model, the decompensation theory, together with several other authors, have pointed out the effects of protective factors that mitigate the impact of these stressors on health [21]. Most protective factors stem from the establishment of personal support networks (family, friendships, or activism) and professional support (access to LGBTQ-affirmative counselling and care services), which enable collective resilience to foster coping skills, self-care, self-efficacy, self-acceptance, and social support [28,29,30].

### LGBTQ-Affirmative Therapy as an Effective Treatment

As mentioned above, the LGBTQ population has been and continues to be stigmatized and discriminated against by society at large [31]. Psychological interventions have also reflected this marginalization and pathologization of LGBTQ people, through the generation of diagnoses in DSM, at least until DSM-III in the case of homosexuality, replacing it with the pathologization of trans* people until today [32]. Currently, this pathologization is perpetuated by some psychologists through the application of so-called ‘reparative’ or ‘conversion therapies’ that are unethical, unprofessional, and potentially dangerous [26]. Additionally, psychotherapeutic approaches fail to consider the idiosyncrasies of LGBTQ individuals and adopt an evasive or non-affirming perspective [33,34]. Fortunately, due to social progress, the push for depathologization has created interest in the scientific community to develop tools and treatments that specifically help the LGTBQ community, such as LGBTQ-affirmative therapy approaches [35].

Historically, the term ‘gay-affirmative therapy’ was coined by Malyon [36] in an article stressing that being targeted by homophobic attitudes, instead of being homosexual, is the cause for the differences in health between gay people and the general population. This term has evolved over the years to become an inclusive therapy for all LGBTQ people. The American Psychological Association’s (APA) guide to affirmative psychological therapy points to the role of stigma and oppression both in daily life and in psychological practice. This guide highlights five fundamental axes to consider in psychological treatment with LGBTQ people: (1) foundational knowledge and awareness; (2) the impact of stigma, discrimination, and sexual minority stress; (3) relationships and family; (4) education and vocational issues; and (5) professional education, training, and research [37].

In addition, the LGBTQ community regularly uses social media and the internet as a gateway to access information about sexual health, identity, etc., [38]. LGBTQ young people tend to perceive online spaces as safer than real life [39]. For this reason, they need to find access to quality content that will help them with their health [40]. The promotion of online programs has emerged over the last decade and increased during the COVID-19 pandemic, allowing people who are geographically distant to work together [41]. Research shows that there are similar levels of satisfaction for both therapists and clients in face-to-face and online therapy [41,42,43].

Therefore, taking these aspects into account, an effective program should activate protective [14] and compensatory mechanisms, reduce vulnerability to stressors [16], and bring about changes in attitudes, as homophobia and stigma are attitudes. Practical conditions for such interventions include action-oriented approaches, participant involvement, and continuous self-validation [44]. Interventions must be tailored to the specific needs and resources of the LGBTQ community [45], incorporating affirmative therapy to empower individuals and their communities, particularly in contexts resistant to diversity [46].

Studies of LGBTQ psycho-educational programs for the general population are scarce. These studies tend to focus on adolescents and young people, although there are also some studies on the training of professionals. In the Portuguese context, we found hardly any data on this type of program. The goal of this research is to understand the effects of participating in an online psychoeducation program, designed for LGBTQ individuals, following the good practices mentioned above, based on qualitative data obtained with its participants. The research question is how did participants experience this program? Accessing their experiences, we will be able to continue improving this kind of intervention, giving agency to the persons who are their main beneficiaries.

## 2. Materials and Methods

### 2.1. Design

As this is an intervention with LGBTQ people, it is necessary for practitioners and the design of the intervention to consider the specificities and resources available to this community [44]. In addition, since group contact can help reduce stereotypes and combat feelings of isolation and invisibility [47,48], a group setting was chosen. Pereira et al. present a synoptic table of eight online interventions for this population. One of these is the Australian Government Department of Health and Ageing’s Mental Health Online model [49]. The program includes 8 weekly sessions on relaxation and mindfulness exercises, identifying disruptive behaviors and maladaptive beliefs, adopting helpful behaviors, building positive identity self-affirmations, and paying attention to self-care (such as diet, sleep, self-esteem, and self-worth). It also includes lessons on setting healthy goals and activating support plans. The authors highlight several improvements achieved by their program, including promoting the health and well-being of LGBTQ people [50]. For the present work, an adaptation of this program was carried out. For this reason, this paper focuses on understanding the experience of participating in an online intervention for the groups, with 8 weekly sessions, using some of the aforementioned techniques and adding strategies for creating social support, like incentives for engaging in peer group activities, the informative literature about LGBTQ identities and developmental processes to strengthen a sense of identity, deliberate exposure to video LGBTQ testimonials and fiction, some specific law knowledge, and literacy in suitable psychological help seeking. The program consisted of various topics such as information about specific associations and collectives, the literature on (affirmative) psychotherapy, health and self-care, video testimonies shown during the sessions, gender and sex differences and explanations, or relaxation exercises [51].

The reflective interviews are the source of material for evaluating evidence of the program’s effectiveness, as well as for collecting suggestions and criticisms from participants. Given the openness of some questions and some directness of others, they are considered semi-structured interviews [52]. The interviews started with an introduction and icebreaker, and went on with the topic questions (process experience; outcomes and content specific for future improvements) and then closure. To learn more about the script, see Appendix A.

### 2.2. Participants

The sample was purposive and voluntary. The decision to use a purposive sample was driven by the interest in including self-identified LGBTQ individuals who have socialized as such. There were 2 interventions with 27 and 11 people each. From the first, 11 participants, and from the second, 5 participants volunteered for the interviews, totaling 16 interviews in the sample. The participants were from different regions in Portugal and between 19 and 45 years old. As it was a blind study, an irreversible term was assigned to each participant (Participant 1, Participant 2, …). Their distribution in classes can be seen in Table 1 and geographic distribution in Figure 1.

### 2.3. Procedure

The present study was carried out in Portugal, and the Ethics Council of the university institute approved the ethical form submitted by the research team. The participants were recruited through an LGBTQ association in Lisbon. The association promoted the program on their social media with a link for enrolment. The psychoeducator and main researcher of the program was a 43-year-old, gay, Caucasian, male psychologist.

All participants had a preliminary individual meeting with the psychoeducator, where the concept of the program was presented and their fitness for enrolment checked. The exclusion criteria were not being LGBTQ-identified; being underage; being visibly unstable in the preliminary interview (e.g., showing thought or speech incoherence, or too many paralinguistic incidents indicating high anxiety, the activation of body movement other than that coherent to narrative expression, or an absolute eye contact intolerance); or stating having very high social anxiety. Any questions people might have had were answered. After this, those who were to proceed were asked to sign a form that had both the informed consent and the link for attending the sessions.

The next phase of the project was the completion of questionnaires, which was carried out before the start of the intervention, as well as after the last session of each program. The first intervention consisted of 9 sessions (one session per week), in three different classes. The second intervention started two months after the first ended with one class receiving 8 sessions. The two interventions were carried out with different people. The program had the same content for all groups and both interventions, but the time each group spent sharing on each topic was flexible. This design (based on the minority stress model of social support as a protective mechanism) was intended to provide the experience of sharing amongst peers and whether it differed from doing it in a large queer group or more specific orientation/gender-identity-focused groups (not part of this study). One day a week, for 9 and 8 weeks, respectively, there was an online group meeting where the activities were presented, and participants could listen to each other’s experiences.

As the principal investigator also led the sessions, anonymity had to be created to ensure the freedom of feedback. Twelve external psychologists were invited to conduct the reflection interviews, our main source of data. The invited external psychologists were volunteers from psychology departments who were aware of the study population and topics, plus a senior researcher and one of the authors of this paper. Six identified as male and the others as female. The age ranged from 23 to 53 years. The training consisted of two 1.5 h sessions designed to raise awareness of everyone’s inevitable homo-bi-trans-phobic cultural biases. Role plays on gender neutral language were used. They were instructed to maintain an attitude of respectful curiosity (interested, but not prurient) and to follow the lead of the participants. Finally, another role play was conducted using the note-taking reflective interview method to practice the understanding stance of the exercise. The script focused on themes like protective mechanisms (and compensation resources) that might have been set in motion; attitude changes in an acceptance direction; and if the participants connected the program’s components with personal changes that might have occurred (Appendix A). Afterwards, there was a concluding interview with the principal investigator. In this interview, the principal investigator talked with each participant about all aspects that were worked on and reported the scores on questionnaires that were completed both before the start and at the end of the program, asking how they reflect their experience (Appendix B).

It should be noted that the data presented here were not derived from transcribed answers, as the interviews were not recorded. Instead, notes were taken during the interview using pedagogical techniques to assess the understanding of a narrative [53]. Interviewers were instructed to always ask if the note fully grasped what the participant wanted to say about each topic. Through the articulation of this shared dialogue process, meanings were refined, and knowledge was produced through reflection [54]. Thus, the notes taken from each answer were returned to the interviewee so that they could confirm, adjust, reflect on them, and propose a formulation until the final note was reached. This process aimed to create a context free of defenses or anxiety and, it is hoped, make it possible for the answers to be closer to the participants’ truth. These notes are the data that were analyzed. This technique was selected with the aim of creating an atmosphere where participants would feel understood (by the interviewer) and that their actual experience in the program really mattered. This is intended to be a restorative experience for people in this population who, as a minority, often feel misunderstood. There was a concluding interview after this one to verify if this purpose was accomplished, but it is outside the scope of this paper.

### 2.4. Data Analysis

A data analysis was carried out with the qualitative analysis software QDA MINER lite V2.0.9. A content analysis was used to deepen and analyze the data [55]. The content analysis was carried out in several stages.

First, initial contact with the data was made, via fluctuating reading [56], creating terms to the first coding cycle, where provisional coding was crafted in a top-down fashion, organizing the data in two main dimensions, based on the literature and the researcher’s field experience [57].

Second, repeated adjustments were made to make the codes more specific and avoid overlapping them. This second phase of coding consisted of an inductive approach [58]. During this phase, the operational definitions of all codes were created and organized within the dimensions according to Bardin’s exclusion criteria [56]. The dimensions are (1) protective mechanisms (from the minority stress model, with two categories: (a) coping skills and (b) social support) and (2) attitudinal changes (from the decompensation theory, contact hypothesis, parasocial contact hypothesis, and social stigma literature).

A sample of the final codes was coded again, days apart, to check the degree of agreement between each round [58]. The intra-coder agreement was close to 85%, so this codebook was used. Figure 2 shows a schematic of the steps that were followed. For example, the definition of the code ‘coping skills (information)’ is ‘indications of use of information as a coping resource (cognitive restructuring, framing, normalization)’ or, in the case of the code ‘social support-group integration’, which is defined as ‘signs of integration into groups or activities with an identity component’. More information on the codes used can be found online [51].

## 3. Results

All cases show evidence of the variation and improvement in protective mechanisms and indications or variations in attitudinal changes. However, it is important to note that there is variability among participants, with some participants showing less variation in the impact of the program. Figure 3 and Figure 4 show the incidence of cases mentioning each of the codes in the interviews.

### 3.1. Protective Mechanisms of Minority Stress Model Dimension

#### 3.1.1. Coping Skills Category

The coping skills category refers to individual skills that have been developed or improved during the program. Within this category there are four codes: (1) information as a cognitive coping resource; (2) behavior changes; (3) resolution behaviors; and (4) personal resources.

Within the first category, in the code—with information as a cognitive coping resource—we can see how participants have used the information gained from the program as a resource to understand and improve their knowledge about the identity group they belong to and as a cognitive reframing mechanism to normalize or accept themselves better. For example, Participant 1 says how important it was for him to be aware that internalized homophobia is not just a personal product, but a socially constructed one: “*The issue of internalized homophobia, realizing that this is much bigger than us. Realizing that I’m not the only one with this blockage. It was good*.”. This may have released the guilt that this person may have felt due to the discriminations suffered throughout his life.

Related to this, Participant 2 discusses how learning about the experiences of the LGBTQ community in general also gave her a better understanding of her own experiences: “*It helped me to better understand things I’ve lived through, particularly at Christmas time. I felt very represented by what we said about those times.*”.

This knowledge helped participants to know more about themselves and make sense of their experiences as being usual ones within the LGBTQ community, thus changing a script of isolation. Also, it raised awareness to the need of knowing more about LGBTQ people and to be involved with the community: “*It increased my realization that we need to know more about specific aspects of the LGBT community, we need to get more involved*” (Participant 3).

The second code is behavioral change during the program. Participant 3, for example, comments on how he has started to express himself more at work: “*I’ve started to feel more comfortable talking about certain situations I observe at work*.”. Participant 4 started wearing LGBTQ symbols: “*Another very important thing I changed was to start using symbols, with the flag! I didn’t do that out of fear.*”. These participants seem to have gained agency and confidence over the course of the program, mobilizing them to change their behavior.

Through psychoeducation on affirmative psychotherapy, some participants have become aware of which professionals they would feel comfortable working with on their mental health. Some were more aware of the need to improve their mental health, which can be included in the code—a resolution for starting more adaptive behaviors. To achieve this goal, some opted for inner listening: “*It’s no longer the search for an object, it’s the realization that something is wrong and I’m the one who needs to go after it. I became more interested in the search for what I really need*” (Participant 5). Others, for affirmative therapy: “*I realized that I needed an affirmative LGBT psychotherapy* because *my mental health was very weakened*” (Participant 4).

Finally, some people discovered other resources or improved existing ones that they can apply in their lives: “*Mindfulness, in which they were asked to close their eyes and imagine themselves in a place where there was a ball. In this ball you would put all your most dubious feelings, and then you would feel the ball*” (Participant 6).

#### 3.1.2. Social Support Category

This category is related to perceived social support, which is necessary for mental health. The first code obtained is related to the experience of participation in the group itself. Being part of a group allows you to get away from the feeling of loneliness and feel more supported. Also, because it is a confidential space, it allows people to feel that it is a safe space. Most participants commented that it was a positive experience, as in the case of Participant 1: “*Having this group experience has given me more security, more confidence*”; or Participant 7: “*that the group was a moment for this, to be focused and to feel safe and supported in our reflections*”.

The following code refers to parasocial contact due to the presentation of several videos with testimonies of LGBTQ people. Thus, not only direct contact with people, but through indirect contact, stereotypes towards one’s own community are also reduced. These two codes refer to Allport’s contact hypothesis as well as to the parasocial contact hypothesis [47,59]. The participants reported that watching these videos gave them different perspectives: “*Watching the video. Visualizing several videos: the story of the trans boy, from a totally different perspective*” (Participant 4).

In addition to in-group implications, participants highlighted that they had perceived changes in general social support, as is the case of Participant 8: “*I felt more included, more accepted and less afraid of being who I am and relating to other people. More integrated*”. The participants also felt more active in the LGBTQ community through participation in the program: “*Seeking to be more active in the community*” (Participant 2). Finally, within the code of social support, some people were able to communicate better with their families thanks to the program, like Participant 1: “*The problem of family issues. I needed to hear examples of strategies for dealing with the family. It ended up helping me to talk to my mum*”.

### 3.2. Attitudinal Changes Dimension

The second dimension is attitudinal changes. A category with three codes, valence, identity integration, and visibility, used to elaborate on attitudinal changes. The valence code is related to the most positive changes in attitudes, both their own and towards the people of the LGBTQ community. Several examples can be found. For example, Participant 1 feels more empathetic towards LGBTQ people who do not belong to his identity group, as well as more empowered:

“*Perhaps I feel a little more relaxed when I meet LGBTI people who don’t belong to my close group (I’m realizing this now, as if there’s a different empathy, more empathy). […] I’m more unblocked, in an affective sense. I feel more informed, more empowered, because this information is more validated, not so isolated. It strengthens me.*”

Participant 3, for his part, feels more sensitive to the cause and Participant 9 feels that her relationship with herself has improved:

“*I’ve become a bit more sensitive to the cause*”(Participant 3)

“*Being better to myself, not being so demanding and critical*”(Participant 9)

The identity code is related to the integration of a minority identity into personal identity, as well as the integration of LGBTQ identity development or identification with referents. The participants addressed the experience of identity as: “*Realize some things and realize that the way we construct our identity shapes the way we deal with rejection, existential dilemmas, with coming out*” (Participant 3). They also commented on their experience by revisiting their history: “*I ended up realizing that I’ve always been like this, I just didn’t understand why and now I can see the difference. I finally see a logic. I feel I’ve found myself more on the identity issue*” (Participant 10). “*Clarifying things about the past or showing how much I don’t know about the past and understanding certain concepts*” (Participant 11). Some participants compared the past and the present: “*This exercise allowed me to see that some things have changed a lot and others not so much*” (Participant 12).

The last code analyzed is visibility. It is related to less concealment or more expression as an LGBTQ person. Some participants focused on how they expressed themselves in the past, such as Participant 6: “*I used to feel that I hid a lot of my more feminine expression. It was very confusing for me*”. Others focused on how they express themselves now: “*more comfortable expressing myself (...) more secure in expressing my sexuality*” (Participant 3). Or, they focused on the change in their emotions, such as Participant 8: “*Today I don’t hide, I’m not afraid*”.

## 4. Discussion

Several online programs with this population have been studied to assess their effectiveness in reducing symptoms such as substance abuse, depression, anxiety, or sexual risk behaviors; however, few have studied the improvement of subjective well-being in general [60]. The present work focuses on understanding the experience of taking part in a LGBTQ-affirmative psychoeducation online group program. According to the results, we can say that the program has been experienced as beneficial by the LGBTQ people who participated in it. The online group format was relevant, as it reduced the feeling of isolation and fostered contact with other LGBTQ people. This also allowed for a naturalization of shared experiences, also addressed as the universality factor [61].

However, looking at Figure 1 and Figure 2, we can see that not all aspects addressed in the program have been equally important for all participants. If we look at the data according to the categories generated, the category that has brought about the most changes overall and worked best for most of the participants is the change in attitudes, as all three codes in it have a high incidence. These data align with the literature, which discusses the effectiveness of affirmative approaches with individuals who have high internalized homophobia. In other words, negative attitudes towards oneself and one’s own group can be changed through affirmative approaches [62]. Furthermore, these affirmative approaches will reduce the proximal stressors generated by internal self-stigmatization.

The next most prevalent category was coping skills. Several studies highlight the importance of good coping skills for greater well-being and less risky behaviors, as shown in several systematic reviews [28,29,30]. In this category, we find the code ‘information’, which refers to knowledge shared in the program sessions. This code is present in 100% of cases, confirming the importance of including information in LGBTQ programs so that people can understand themselves and their group better [63]. Being able to access information that they have never had before is relevant to improving their self-awareness and, therefore, developing better coping skills. In addition, it is a program in which they have been provided with resources and have been able to explore for themselves those that best fit their way of being, thinking, or feeling. In this way, they are given options to choose which strategies they want to use in the future.

In addition, among the behaviors acquired, some participants mentioned the relaxation proceedings and exercises that took place in every session as a strategy they started doing on their own, and that helped them cope better with daily problems. These results are consistent with the existing literature, which explains how relaxation and mindfulness are effective tools that improve emotional regulation and expression, and problem solving, among other aspects [64,65]. Therefore, creating safe spaces and developing coping skills allows for the further development of agency. This can improve people’s mental health, as other studies have shown [16,66]

Finally, the category of the lowest incidence was the codes for family and a general feeling of integration. The literature exposes that LGBTQ people tend to have less family support and less support in general [14]. Therefore, we can say that the present data are consistent with the existing literature. It should also be noted that the intervention only took place with LGBTQ people, not with their families or the general population. Nevertheless, some participants mention improvements in their relationship with their family and how the program enabled them to talk to some family members. In the same category, there are codes indicating better outcomes such as the group experience of psychoeducation or the feeling of being integrated in the group. This is supported by the existing literature which explains the importance of the existence of safe spaces [67]. Viewing LGBTQ testimonies in video format (indirect) created a beneficial social support environment for the participants, which they feel has improved their well-being. The existing literature explains that social media has become a space to visibilize LGBTQ people, reducing stereotypes and showcasing the diversity of LGBTQ lives [68].

Therefore, we can see that there is variability in the personal experiences of the participants. We must consider that each person started from a different personal context. Some people may have more personal resources, so the program has not made much difference in this respect. Others may be more resistant to change. In terms of social support, people may respond differently for different reasons: they may feel perfectly integrated at the moment, they may be in contexts where integration is difficult, or they may even need to improve their social skills. Finally, attitudes will vary or not, depending on their previous attitudes. The data show us the importance of considering group psychoeducation as a useful way to work on the empowerment and wellbeing of the LGBTQ population. It is also important to continue working with and improving this type of program. Listening to the participants and their suggestions is a feedback resource for the constant improvement of the protocol of this kind of program. However, we must always consider the characteristics of each group to be able to make small adjustments based on their needs.

## 5. Conclusions

This study explores the participants’ experiences on a psychoeducation program carried out in Portugal, aimed at enhancing psychological well-being and promoting coping skills across diverse populations. The findings provide preliminary evidence of the positive impact of such interventions within the Portuguese cultural context, leading to several key conclusions.

Firstly, the participants’ experience has been positive, increasing their knowledge about the community they belong to, improving their coping skills, and increasing their sense of belonging and social support. Secondly, the design and delivery of the program, tailored to Portuguese cultural values, has played a crucial role in its success. The inclusion of practical examples and group dynamics has aligned with local norms, facilitating higher acceptance and engagement from participants. Finally, participants seem to have changed their attitudes towards themselves and other LGBTQ people, becoming more aware of the challenges they may face and valuing the changes they have made throughout the program.

In addition, running the program online allowed it to reach more people, as the participants lived in different municipalities and did not have to travel to a distant location, but instead took the sessions from a comfortable place.

In conclusion, the results are encouraging, and the program seems to be beneficial for the LGBTQ population and for the participating groups. We can highlight the importance of psychoeducation as a key process for them to better understand themselves and their own life story from a more complex perspective. Psychoeducation is, therefore, an effective and holistic support for a community that is heterogeneous and has different health and well-being needs. Both the participants and the institution expressed their desire to continue the program, which testifies to its success [51].

## 6. Limitations and Future Research

This work is not without limitations. Firstly, the sample is too small to allow the generalization of the findings. In addition, as the interviews were not recorded, the relevant data may have been lost. To avoid biases, the collected data and results were shared with the participants. However, in the subsequent discussion of the data, the participants were only informed. In this way, it is relevant to consider the described characteristics of the researchers, as they establish a situated point of view that conditions the generated knowledge [69,70].

Affirmative psychoeducational programs have proven to be an effective tool for enhancing the well-being of the LGBTQ population by strengthening protective factors and mitigating internalized negative attitudes [53]. Additionally, their online format offers a significant advantage by increasing accessibility in contexts where cisheterodissident realities are marginalized due to geographical, social, or cultural barriers [50].

In this regard, it is essential to expand the scope of these programs to include not only LGBTQ individuals, but also their families and individuals with negative attitudes toward this population [28]. Such an approach aims to foster the creation of environments with reduced proximal stressors, promoting safer and more affirmative spaces. Future programs should also adopt a personalized approach tailored to the specific needs of each subgroup within the LGBTQ spectrum, considering the unique challenges faced by lesbian, gay, bisexual, trans*, intersex, asexual, and other identities [28]. In addition, future studies should involve more participants to corroborate these findings.

The implementation of longitudinal studies of online programs should be prioritized in this field, as they would allow for a deeper and more sustained analysis of the impact of interventions over time, providing a more comprehensive understanding of the processes of change [71]. In carrying out these studies, it is important to consider the three broad themes suggested (coping skills, social support, and attitudes). Furthermore, this approach offers professionals key tools to more effectively and sensitively support and accompany these realities.

## Figures and Tables

**Figure 1 healthcare-13-00115-f001:**
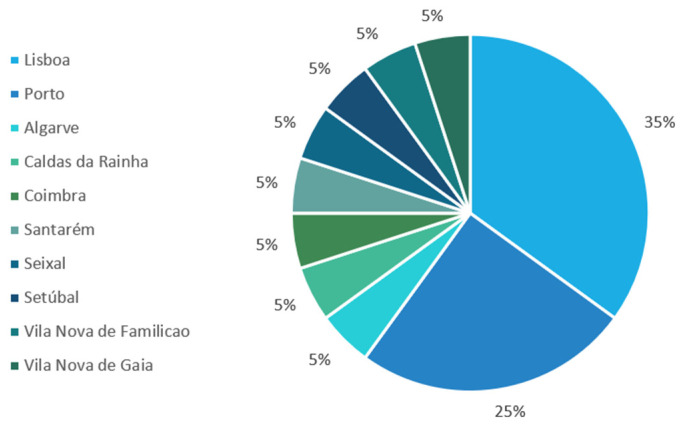
Geographical distribution.

**Figure 2 healthcare-13-00115-f002:**
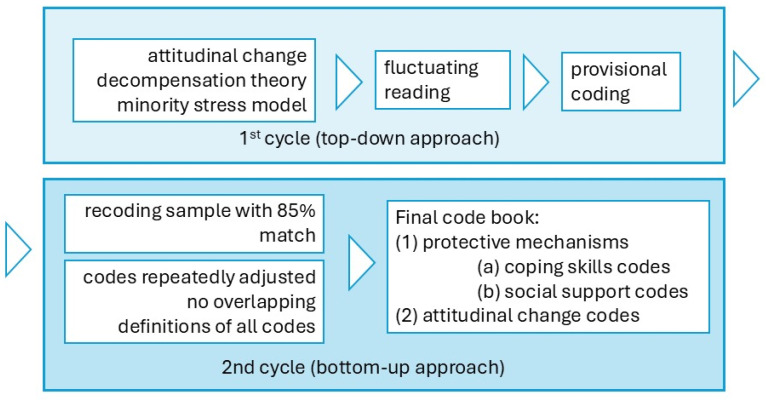
Data analysis outline.

**Figure 3 healthcare-13-00115-f003:**
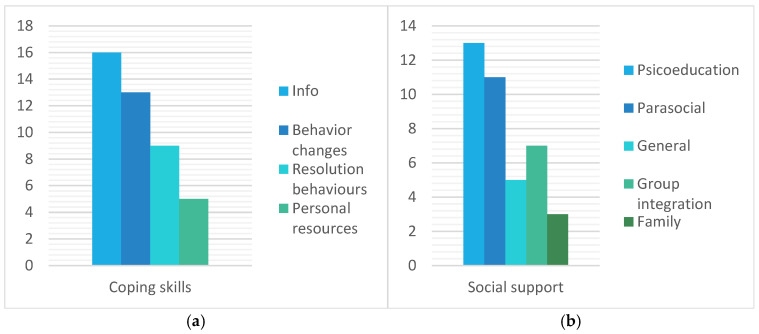
Incidence of cases in the protection mechanisms of the minority stress model dimension. Divided into categories: (**a**) coping skills and (**b**) social support.

**Figure 4 healthcare-13-00115-f004:**
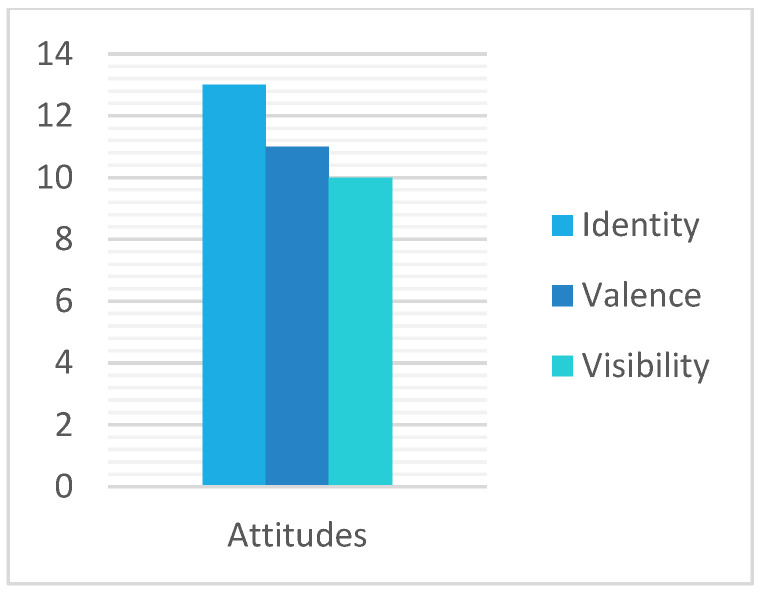
Incidence of cases in the attitudinal changes dimension.

**Table 1 healthcare-13-00115-t001:** Distribution of the sample in groups.

Intervention	Groups	Focus	Total Participants	Interview Participants
Intervention 1	Class 1	Sexual orientation	11	11
	Class 2	Sexual orientation	10	
	Class 3	Gender identity	6	
Intervention 2	Class 1	LGBTQ	16	5

## Data Availability

Data are contained within the article.

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
