# Peer review of "Voices from the Rainbow: Exploring Participants’ Experiences in an Online LGBTIQ+-Affirmative Psychoeducation Program"

_healthcare, 2025, doi:10.3390/healthcare13020115_

Round 1

Reviewer 1 Report

Comments and Suggestions for Authors

Dear Authors,

Thank you for providing me with the opportunity to read this interesting manuscript. Below I have listed some of my comments:

1) Make sure to stick to Americal spelling. In the abstract and throughout the paper, the word program is written either as programme or program. Therefore, try to be consistent.

2) The introduction would benefit from an expansion on online therapy/programs. Since this is the focus of the study, it would have been nice to ptovide some context.

3) The last paragraph of the introduction that states the objective can be enriched by linking gaps to the objective and highighting the key implications.

4) While some aspects of the intervention are mentioned, there is insufficient detail on how these were implemented in the sessions. Did participants have to watch pre-recorded videos or were they live sessions? How were peer group activities facilitated in this online setting?

5) Were interviewers mental health professionals or laypeople?

6) The absence of recorded interviews raises concerns about the potential loss of nuance in participants’ responses. Although the iterative note-taking process is described, it lacks clarity on how biases (e.g., selective note-taking) were mitigated.

7) It would be nice to provide more information such as race and economic background about your sample aside from age and geographic distribution so that the findings can be more generalizable.

8) being visibly unstable as a criterion is subjective. How were people evaluated based on this criterion?

9) The creation of operational definitions is mentioned, but no examples are provided. Consider including one or two sample definitions to illustrate how they contributed to the rigor of the coding process.

10) For the findings section, rather than being semantic and use descriptive language, try to go a little deeper into that the participants' accounts actually mean (latent analysis).

11) The discussion acknowledges that the program’s impact varied among participants but does not delve deeply into why.

12) The recommendation to conduct longitudinal studies is excellent, but need to expand on this. What kind of longitudinal studies? online or offline programs? what kind of intervention components?

I hope this feedback is helpful.

Author Response

We sincerely appreciate the insightful comments provided, as they have greatly contributed to enhancing and improving the manuscript. In the attached file, you will find a detailed, point-by-point response to each of the comments, along with a description of the revisions made to the text. Additionally, all changes have been marked in red within the manuscript, which we have uploaded to the platform for your review.

Reviewer 2 Report

Comments and Suggestions for Authors

Dear authors,

First of all, I would like to congratulate you on your study. It is very interesting in that it highlights the importance of addressing the mental health of LGBTQ people through psychoeducation programs.

Regarding the reviews of your manuscript, I have found that the work is well founded and each of its sections is presented clearly and correctly. However, the following recommendations should be made as suggestions for improvement:

- In Figure 1, the borders should be removed, so that adequate quality is presented.

- In the Limitations of the study section, it should be noted that one of the limitations has been the limited number of participants. Therefore, future studies should increase the sample size to corroborate these findings.

- It is necessary to present the conclusions section at the end of the discussion, in which the most relevant findings of the study are inserted. This will allow the reader to more easily read the conclusions of the study.

Author Response

(The authors gave the same response as above.)

Reviewer 3 Report

Comments and Suggestions for Authors

The purpose of this manuscript was to understand the effects of participating in an online affirmative programme for LGBTQ people. The focus of the study fits within the scope of the journal and is of relevance to its readership. Below, I note specific areas to strengthen the manuscript and note concerns about the methodology described. Overall, the manuscript should be reviewed for grammatical errors/clarity throughout. I wish the authors the best of luck as they continue to move this research forward.

Introduction

·         The introduction is thorough and does a good job of describing the etiology of mental health among LGBTQ people.

·         There is extremely little information about the interventions that are being evaluated and how the concepts (i.e., protective factors) described in the intervention are covered in the interventions.

Methods

·         There appears to be some information about one of the interventions (Australian Government Department of Health and Ageing's Mental Health Online model), but it is unclear if this is the exact one used or if it is an adaptation.

·         Moreover, there isn’t information on a second intervention

·         It is unclear what the “Groups” in Table 1 are in reference to and how the programming for these seemingly different groups varied.

·         The authors state that “As the principal investigator also gave the classes, we had to create anonymity to ensure freedom of feedback” but then state that “Afterwards, there was a concluding interview with the principal investigator (Appendix B).”  It is unclear how freedom of feedback was ensured.

·         I have not previously heard of the procedure described for analysis. It seems as though this may be subject to interviewer bias. Can the authors please describe characteristics of the interviewers and any potential biases that may have impacted analysis? Was there any reflexivity/bracketing that may have occurred prior to the interviews and/or coding?

Results

·         The results were presented clearly regarding protective mechanisms. However, I am wondering if there were any findings related to improved mental health (or, if they are reported elsewhere).

Discussion

·         Given the setting (Portugal) it is curious that there was no mention of cultural values/beliefs and how they may impact mental health.

Comments on the Quality of English Language

Overall, the manuscript should be reviewed for grammatical errors/clarity throughout. 

Author Response

(The authors gave the same response as above.)

Round 2

Reviewer 1 Report

Comments and Suggestions for Authors

Dear Authors,

Thank you for your responses. The article now is enhanced and can be published.

Author Response

Thank you for your feedback and for your positive assessment of the revised article. We are glad to hear that the improvements have enhanced the manuscript and that it is now ready for publication.

Reviewer 3 Report

Comments and Suggestions for Authors

The authors have addressed the majority of the comments. Although the authors have described characteristics of the interviewers, they still have not addressed any potential biases that may have impacted analysis.

Comments on the Quality of English Language

The manuscript should be further reviewed for grammatical errors/clarity throughout.

Author Response

Thank you for your valuable feedback. We have now added the limitations you pointed out, marked in blue for clarity. Additionally, the manuscript has been reviewed by a language expert, and we have made small revisions to improve the quality of the English language and ensure greater clarity.

We hope these changes address your concerns.